# Purification and Characterization of a Novel Antifungal Flagellin Protein from Endophyte *Bacillus methylotrophicus* NJ13 against *Ilyonectria robusta*

**DOI:** 10.3390/microorganisms7120605

**Published:** 2019-11-22

**Authors:** Yun Jiang, Chao Ran, Lin Chen, Wang Yin, Yang Liu, Changqing Chen, Jie Gao

**Affiliations:** 1College of Life Science, Jilin Agricultural University, Changchun 130118, China; jyjyccq@163.com (Y.J.); 15843020543@163.com (C.R.); a308269630@163.com (L.C.); yinwang2010@163.com (W.Y.); 2College of Plant Protection, Jilin Agricultural University, Changchun 130118, China; y_liu10@jlau.edu.cn (Y.L.); jiegao115@163.com (J.G.)

**Keywords:** endophyte *Bacillus methylotrophicus*, antifungal activity, flagellin, *Ilyonectria robusta*, biocontrol agents

## Abstract

Endophyte *Bacillus methylotrophicus* NJ13 was isolated from *Panax ginseng*. Its sterile fermentation liquid showed a significant inhibitory effect against *Ilyonectria robusta*, causing the rusty root rot of *P. ginseng* and *P. quinquefolius*. The antifungal protein was obtained after precipitation by 20% saturated ammonium sulfate, desalted by Sephadex G-25, weak anion exchange chromatography, and gel filtration chromatography. SDS-PAGE showed that the purified protein was approximately 29 KDa. The antifungal protein after desalting was not resistant to temperatures higher than 100 °C, resistant to acid conditions, and did not tolerate organic solvents and protease K. The amino acid sequence of purified antifungal protein had an identity of 76% to flagellin from *Bacillus velezensis*. The isoelectric point of the protein was 4.97 and its molecular mass was 27 KDa. Therefore, a specific primer G1 was designed based on the flagellin gene sequence, and a 770 bp gene sequence was cloned in NJ13 genomic DNA, which shared the same size of flagellin. There were ten base differences between the gene sequences of flagellin and the cloned gene, however, the amino acid sequence encoded by the cloned gene was identical to the flagellin. In conclusion, the antifungal protein produced by biocontrol agent NJ13 contained a flagellin protein.

## 1. Introduction

*Panax ginseng* is a perennial herb that has been used as traditional medicine for thousands of years because of its antifatigue [1], anticancer [2], and immunity-enhancing properties [3]. During five or six years of cultivation, ginseng plants may suffer from several diseases that can lead to crop losses and reduction of root quality. Rust rot is one of the most serious soil-borne diseases as it can reduce the yield of *P. ginseng* and seriously restricts the economic development of *P. ginseng* [4,5]. *Ilyonectria robusta* is an important pathogen for the rusty root rot of *Panax ginseng* and *Panax quinquefolius*, and previous studies have reported some strains, which mainly included *Trichoderma*, *Actinomyces*, and *Bacillus*, exhibited antagonistic effects on *I. robusta* [6,7,8]. However, there are few reports on the isolation and purification of the compounds produced by the biocontrol agents against *I. robusta*. Our previous work showed that the fermentation broth of NJ13, which was a strain of *Bacillus methylotrophicus*, contained antifungal proteins that could inhibit the growth of *I. robusta*. The crude proteins obtained by ammonium sulfate precipitation method could exhibit a 62% inhibition rate against the spore germination of *I. robusta* with a low concentration of 31.25 mg/L [9]. Hence, it could contribute to fight rust rot disease and improve the economic development of *Panax ginseng* through identifying protein which could inhibit *I. robusta*. In this study, antifungal proteins of NJ13 were isolated and purified by ion-exchange chromatography, gel filtration chromatography, electrophoresis, and the other methods. The proteins were identified by mass spectrometry, and the sequence and function of the proteins and related information were inferred by it.

## 2. Materials and Methods

### 2.1. Isolation and Cultivation of the Strain

The strain NJ13 used in this study was kept in our laboratory. This strain was isolated from the stem of *Panax ginseng* collected in Changchun, Jilin province, and identified to be *B. methylotrophicus* [10]. The target pathogen used in this study, *I. robusta*, was conserved in the Plant Pathology Laboratory of Jilin Agricultural University. NJ13 was cultivated in 75 mL medium within a 250 mL flask which contained purified water, 3% glucose, 1.5% starch, 1.5% yeast extract powder, 0.1% K_2_HPO_4_, and 0.5% NaCl at pH 7.0, at 28 °C, in the shaker with 160 rpm for 72 h.

### 2.2. Extraction of Antifungal Proteins

After cultivating for 72 h, the medium containing NJ3 was centrifuged at 4 °C with 22,200× *g* for 30 min, and cell-free supernatants were collected by 0.45 μm filters to remove debris. Then, 20% saturated (NH_4_)_2_SO_4_ was added into supernatants and incubated overnight at 4 °C. The precipitate was collected by centrifugation at 22,200 g for 15 min. After the precipitate was dissolved in deionized water, the dialysis was carried out with a dialysis bag with a cut-off component of 8–14 KDa in deionized water overnight at 4 °C. Then, BaCl_2_ was added to detect the white precipitate, and the crude proteins in the dialysis bag were collected. Following this, the proteins were concentrated using a vacuum freeze drier at −40 °C for 72 h and the dried material was dissolved in a suitable volume for further analysis.

### 2.3. Antifungal Assay of the Proteins

The antifungal activity of proteins produced by NJ13 was evaluated by disk diffusion assay at filter paper disks [11]. Firstly, 80 μL *I. robusta* conidia suspension (10^5^ conidia/mL) was uniformly coated on the PDA disk. Then, wells were created on the disk, 80 μL proteins solution (1 mg/mL) was added into one well, and 80 μL buffer which was used to dissolve the proteins was used as the control. The plates were incubated at 20–25 °C for 3 days, and the antifungal activity of proteins was determined by measuring the diameters of inhibition zones of *I. robusta* growth.

### 2.4. Toxicity Test of the Proteins

The toxicity of the proteins on *I. robusta* was tested by the mycelium growth rate method. PDA disks which were with protein concentrations of 1.875 × 10^5^ mg/L, 1.875 × 10^4^ mg/L, 1.875 × 10^3^ mg/L, 1.875 × 10^2^ mg/L, 1.875 × 10^1^ mg/L, and 0 mg/L (control group), respectively, and with the same diameters of 9.0 cm, were firstly prepared. Then, agar disks with diameter 5 mm were cut from the edge of PDA which had cultured *I. robusta* for 7 days. After these agar disks were inverted on the prepared PDA disks, they were cultured under 25 °C until the colony diameter in the control group reached up to 60 cm. Each experiment was replicated 3 times. The colony diameter was measured by the cross-crossing method and the inhibition rate was calculated based on the percentage change of the colony diameter compared to the control group. In the end, EC_50_ was calculated based on inhibition rate and the corresponding protein concentrations. [12]. 

### 2.5. Purification and Identification of Activity Proteins

The crude proteins were dissolved in 20 mM Tris-HCl of pH 8.0 (buffer A), and 0.22 μm filters were used for sterilization. This solution was further purified by gel filtration chromatography using a Sephadex G-25 column (GE, 5 mL). The column was thoroughly equilibrated with buffer A firstly, and then 2 mL sample with a mass concentration of 20 mg/mL was loaded. The elution flow rate was 3 mL/min, and each absorption peak of the specific wavelength of 280 nm for proteins was collected by retention time. The antifungal activity of each tube was determined according to the disk diffusion assay method and the eluted unit of each tube was discarded or combined according to the activity and components. The antifungal activity peak was freeze-dried for further analysis.

The proteins, which were desalted and freeze-dried, were dissolved in buffer A (10 mM tris-HCl, pH = 8.0) with a concentration of 20 mg/mL, and were then passed through 0.22 μm filters. After using buffer A to equilibrate the weak anion exchange column (DEAE-Sepharose Fast Flow Column, GE, American) with 5 column volumes, a 2 mL sample was loaded, and the equilibrium and elution flow rates were kept constant at 2 mL/min. After loading, the unbound protein was eluted with buffer A for 5 column volumes, and 10 column volumes of buffer B (1 M NaCl) was used to elute the bound protein with a linear NaCl gradient (0–1 M). Each eluted peak was collected and fully dialyzed by deionized water overnight at 4 °C. After lyophilization, the antifungal activity of each peak was measured and the active proteins were prepared and collected on a large scale.

The active proteins collected by weak anion exchange chromatography (DEAE-Sepharose Fast Flow Column, GE, Shanghai, China) were dissolved in buffer A and passed through 0.22 μm micropore filters. The column was equilibrated with 2 column volumes of buffer A. The sample concentration was 20 mg/mL, the loading volume was 1 mL, and the flow rate was 0.75 mL/min. The eluted peak was collected and fully dialyzed by deionized water overnight at 4 °C. After lyophilization, the antifungal activity of each peak was measured, and the active peak protein was prepared and collected on a large scale.

The purity of the target protein was determined by SDS-PAGE. The electrophoresis was accomplished with a 5% stacking gel and 12% running gel. The electrophoretic band was cut out and the target proteins were identified by matrix-assisted laser desorption ionization time-of-flight mass spectrometry (MALDI-TOF-MS, Shanghai Applied Protein Technology Co., Ltd., Shanghai, China).

### 2.6. Effects of Heat, pH, Ultraviolet, Chloroform, and Enzymes on Antifungal Activity

After desalination, the active peak was selected for stability tests. After vacuum freeze-dried, the substance was dissolved in buffer A with a concentration of 1 mg/mL. The solution of the active protein was exposed to a water bath at 40 °C, 60 °C, 80 °C, 100 °C, and 121 °C (0.1 MPa) for 30 min to analyze the thermal stability. The pH stability was determined by adding different solutions (hydrochloric acid and sodium hydroxide) into the supernatant to regulate pH value to 2.0, 4.0, 6.0, 8.0, 10.0, 12.0, 14.0, and incubating the supernatant overnight at 4 °C. Then the antifungal effect was evaluated after the pH of all samples was adjusted back to neutral. The supernatant was treated with ultraviolet irradiation for 12 h to test the ultraviolet stability. The supernatant was mixed with chloroform in an equal volume and shaken at room temperature for 60 min. After centrifugation, the layers were separated and the aqueous phase was taken. The mixture was allowed to stand at 4 °C overnight, and the antifungal effect was detected after the residual chloroform was volatilized. The enzyme stability was detected by assaying the activity after adding proteinase K to the supernatant with 100 μg/mL final concentration of proteinase K, and incubated at 37 °C for 60 min. 

Three replicates were set for each stability experiment, diameters of the inhibition zones were measured for estimating antifungal activity, and the LSD (least significant difference) method was used for analysis in the DPS data processing system.

### 2.7. Cloning of Genes of Antifungal Proteins 

According to results of antifungal proteins measured by the mass spectrometry, primers, which included the restriction enzyme cutting sites of *BamH I* and *Sal I* and protective bases, were designed by Primer5.0 software with the template of the gene sequence of *Bacillus velezensis* UCMB 5033 flagellin (accession number: CDG29557.1).

After the genomic DNA of NJ13 was extracted, the target genes were amplified by PCR with the primer pair of G1-F (CGCGGATCCCAGGGGGAATTTAACATG) and G1-R (ACGCGTCGACCTCCCTCCTTACCGTTTC). The PCR reaction was performed in 25 μL reaction mixture containing 2 μL of NJ13 genomic DNA, 1 μL G1-F, 1 μL G1-R, 12.5 μL Premix *Taq* DNA polymerase, and 8.5μL sterile deionized water. The PCR cycling protocol consisted of initial denaturation at 95 °C for 5 min, followed by 34 cycles of 95 °C for 30 s, 56 °C for 50 s, and 72 °C for 1 min, and a final elongation step of 72 °C for 10 min. As a negative control, the template DNA was replaced by sterile double-distilled water.

The PCR amplified products were cloned using the pMD^TM^18-T Vector cloning kit manufactured by Takara Bio Co., Ltd. The purified PCR products were ligated to the PMD18-T vector and transformed into competent cells of *E. coli* DH5α cells. The plasmid was extracted for recombinant clonal screening and restriction enzyme analysis, the positive recombinant clones were selected, and the amplified fragments were sequenced. Fragments were sequenced by Sangon Biotech (Shanghai) Co., Ltd., and the sequences were analyzed by DNAMAN (6.0). 

## 3. Results

### 3.1. Activity Assay of Proteins

Proteins present in the supernatant were collected by ammonium sulfate precipitation, but a large number of contaminants remained. The column chromatography was used for further desalination. After the crude proteins were desalted by Sephadex G-25, two protein elution peaks (Figure 1) and one salt peak appeared. Among them, the first and the second peaks had antifungal activity. Hence, the 2F peak of protein was selected for further analysis. 

### 3.2. Purification and Identification of Activity Proteins

The active protein 2F desalted was collected, and four protein flowing peaks and three elution peaks appeared after elution by weak anion exchange chromatography (Appendix A). Among them, peak 2-7F had antifungal activity but its inhibition zone was opaque and low content, while peak 2-6F was the main active peak (Appendix A).

The active protein 2-6F eluted by ion-exchange chromatography was collected, and five major elution peaks appeared after elution by the gel filtration chromatography (Appendix A). Among them, the peak 2-1-1F was the active peak (Appendix A) and its EC_50_ was 144.52 mg/L (Appendix A).

The concentrated antifungal protein 2-1-1F (20 μL) was taken and detected by SDS-PAGE. The results revealed a single monomeric protein band with a molecular mass estimated to be 29 KDa (Figure 2). It was obvious that the protein had high purity and no subunit, and it could be identified by mass spectrometry.

According to the results of primary and secondary mass spectrums (Appendix A), the peptide sequences of the target protein was blast against the database. It was found to have a 76% identity to flagellin of the *B. velezensis* UCMB 5033 (protein ID: S6FXJ3) (Appendix A). The target protein consisted of 258 amino acids with an isoelectric point of 4.97 and a molecular mass of 27 KDa. The apparent molecular mass of the target protein and the antifungal protein measured by the electrophoresis experiment were almost the same, therefore, the purified antifungal protein was initially determined as flagellin.

### 3.3. Effects of Heat, pH, Ultraviolet, Chloroform, and Enzymes

The peak 2F after desalination was selected for the stability study. The results showed that the 2F peak of protein was not resistant to heating at temperatures higher than 100 °C (Appendix A). Antifungal proteins were tolerant to acid conditions but had a greater loss of activity under alkaline conditions of pH 12 (Appendix A) [13,14]. It was not resistant to organic solvents and proteinase K, which resulted in complete loss of its antifungal activity, but it was stable under ultraviolet irradiation conditions (Appendix A).

### 3.4. Detection of Genes of Antifungal Proteins

The NJ13 genome DNA was amplified by PCR with specific primers G1-F/G1-R to obtain a sequence of about 770 bp in length, which was consistent with the size of the target genes in the model strain (Figure 3).

The screened positive clones were subjected to bacterial solution PCR, plasmid sequencing, and double enzyme digestion test. A band of the same size as the target genes could be amplified by bacterial solution PCR. The enzyme digestion analysis showed that the target genes were successfully ligated to the pMD18-T vector and transformed into *E. coli* DH5α cells (Appendix A). The results of the plasmids sequencing showed that the sequence linked to the T vector differed from the target genes by 10 bases (red bases in Appendix A), however, the cloned genes sequences translated by DNAMAN (6.0) were of 100% identity to the flagellin of *B. velezensis* UCMB 5033.

Some flagellin sequences were selected from the NCBI database and compared with the flagellin FlgG sequence of *B. velezensis* UCMB 5033 (Accession number: CDG29557.1). The sequence accession numbers and strain numbers involved were as follows:

Multiple sequence alignment of the above flagellin sequences were performed and a phylogenetic tree was established using the software MEGA-X [15]. After alignment of the amino acid sequence of a protein with NCBI database, it was found that the purified antifungal protein in this study that had 76% identity to the flagellin protein (accession number: CDG29557.1) belonged to the FlgG flagellin family. The phylogenetic tree was divided into three main branches, as shown in Figure 4.

## 4. Discussion

Flagellin is generally used as an adjuvant in immunology to enhance the immune host response level [16,17]. Sun et al. revealed that flagellin conserved N-terminal epitope flg22 was specifically recognized by flagellin-sensitive 2 (FLS2) in *Arabidopsis*. Direct recognition of flg22 by FLS2, which is a leucine-rich repeats receptor kinase, is sufficient for inducing plants’ immune responses [18]. In detail, the recognition of flagellin could lead to rapid phosphorylation of the MAP kinase pathway, and this further generates lots of host-defense proteins that mediate protection against fungal and bacterial infections [19,20]. In certain solanaceous plants, the FLS3 receptor can recognize flgII-28, which is a region of bacterial flagellin that is distinct from the region perceived by the FLS2 receptor, and can enhance immune responses that protect leaf tissues against bacterial colonization [21,22,23]. For the antifungal studies, several flagellin proteins that could antagonize plant pathogens and were sourced from the fermentation broth of biocontrol bacteria *Bacillus* spp. were isolated and purified [24,25]. 

As shown in Table 1, the first branch in the phylogenetic tree contained six sequences composed of amino acid sequences that were similar to the target protein (CDG29557.1), from 100% to 49%. The second branch contained a sequence belonging to the *Bacillus* TasA family (Table 2). The third branch contained five sequences that were flagellin and were reported with antifungal effects (Table 2). These flagellin proteins differed in the number and type of amino acids. The end of the amino acid sequence was conserved, and the middle sequence was disordered and could not be matched (Appendix A). In addition, the sequence similarity to the FlgG family was less than 20%. Hence, it was speculated that such flagellin might have antifungal activity and their antifungal activity might be related to the amino acids at both ends. However, the mechanism of flagellin against phytopathogenic fungi has not been revealed yet. The specific characteristics of the functional groups and spatial structures of the protein could be studied in the future, to lay a foundation for artificial synthesis and heterologous expression.

## 5. Conclusions

The fermentation broth of *B. methylotrophicus* NJ13 was rich in antifungal protein, and a flagellin protein isolated was with a molecular mass of 27 KDa and an isoelectric point of 4.97. According to the results of pre-experiment, it was speculated that the unseparated and unpurified antifungal protein may be a basic protein, which would be isolated and purified in the future study, in order to discover the antifungal-related protein in the strain NJ13.

## Figures and Tables

**Figure 1 microorganisms-07-00605-f001:**
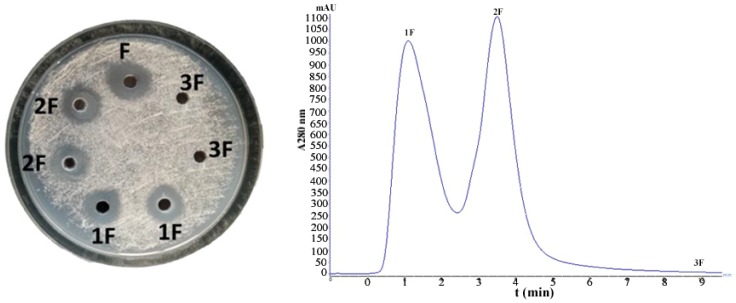
Desalting results of crude protein by Sephadex G-25 and antifungal effect of elution peak (Note: F for crude protein, 1F for peak 1, 2F for peak 2, 3F for peak 3).

**Figure 2 microorganisms-07-00605-f002:**
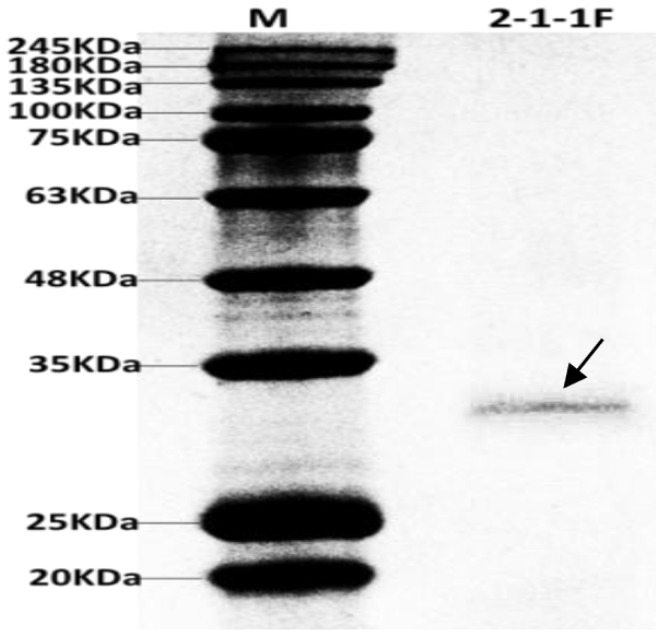
SDS-PAGE of antifungal protein 2-1-1F.

**Figure 3 microorganisms-07-00605-f003:**
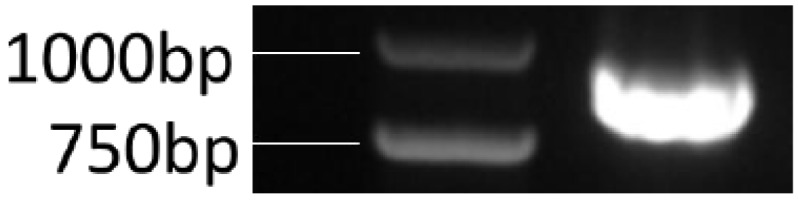
PCR results of the target genes.

**Figure 4 microorganisms-07-00605-f004:**
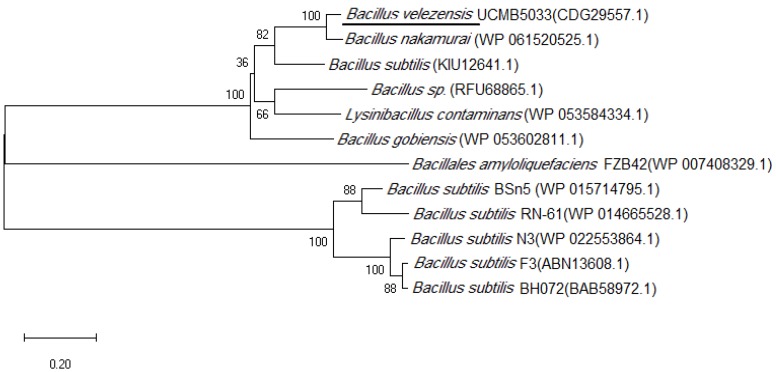
Phylogenetic tree of flagellin.

**Table 1 microorganisms-07-00605-t001:** Sequences with higher similarity to flagellin of *B. velezensis* UCMB 5033 (Accession number: CDG29557.1).

Accession Number	Protein Family	Strain	Similarity
WP 061520525.1	FlgG	*B. nakamurai*	91%
KIU12641.1	FlgG	*B. subtilis*	72%
WP 053602811.1	FlgG	*B. gobiensis*	60%
RFU68865.1	FlgG	*B.* sp. V59.32b	55%
WP 053584334.1	FlgG	*Lysinibacillus contaminans*	49%

**Table 2 microorganisms-07-00605-t002:** Antifungal disease-related flagellin.

Accession Number	Protein Family	Strain and Number
	Antifungal flagellin	
WP 014665528.1	Flagellin	*B. subtilis* RN-61
WP 015714795.1	Flagellin	*B. subtilis* BSn5
WP 022553864.1	Flagellin A	*B. subtilis* N3
BAB58972.1	Flagellin	*B. subtilis* BH072
ABN13608.1	Flagellin	*B. subtilis* F3
WP 007408329.1	TasA	*B. amyloliquefaciens* FZB42

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
