# Peer review of "Purification and Characterization of a Novel Antifungal Flagellin Protein from Endophyte Bacillus methylotrophicus NJ13 against Ilyonectria robusta"

_microorganisms, 2019, doi:10.3390/microorganisms7120605_

Round 1

Reviewer 1 Report

All of my comments have been addressed in the revised manuscript.

Author Response

Dear reviewer:

 Thank you for your comments in this paper.

Yours' 

Changqing Chen

Reviewer 2 Report

The current form of revised manuscript is acceptable for publication in "Microorganisms".

Author Response

Dear reviewer:

 Thanks for your previous comments about this paper.

Yours'

Changqing Chen

Reviewer 3 Report

I think that the manuscript has significantly improved and it can now be accepted for publication after minor revision.

Minor:

1) Line 45: Please do not use "etc" in a sentence.

2) Line 72: After cultivating for 72 h, .....

Author Response

Response to Reviewer 3 Comments

Thanks for your comments and the manuscript has been revised according to these comments.

Point 1: Line 45: Please do not use "etc" in a sentence.

Response 1: This has been revised according to the reviewers’ comment, and the “etc” has been revised to “and the other methods”.

Point 2: Line 72: After cultivating for 72 h, .....

Response 2: This has been revised according to the reviewers’ comment, and the “After cultivated 72 h” has been revised to “After cultivating for 72 h” in the manuscript.

This manuscript is a resubmission of an earlier submission. The following is a list of the peer review reports and author responses from that submission.

Round 1

Reviewer 1 Report

The study identifies an antifungal protein produced by a bacterial endophyte associated with ginseng. The protein appears to be a flagellin that has activity against a rust rot that damages Panax ginseng. The methods and analysis are clear and results support the conclusion that at least a component of the bacterial antifungal activity is attributable to a flagellin secreted by the bacterium. I have a few minor comments:

Line 35: “some strains” is unclear. State the name of the strain.

Line 54: What volume of medium was used?

Line 64: Were these filter paper disks?

Line 74: “peak” associated with absorption of specific wavelength?

Line 87: Sate the type of column used here.

Line 93: Cut out instead of cut “off”

Line 159: Delete “at”

Line 200: Table1 and phylogenetic Figure, and description associated with them, belong to the Results section.

I would suggest including at least one of your supplementary figures showing clearing zones in the main Results section so readers quickly get a clear idea of the assay being employed.

Author Response

Response to Reviewer 1 Comments

Point 1: Line 35: “some strains” is unclear. State the name of the strain.

Response 1: The names of the strains were added in the context “which mainly includes Trichoderma, Actinomyces, and Bacillus”.

Point 2: Line 54: What volume of medium was used?

Response 2: 75 mL medium was used to cultivate NJ13.

Point 3: Line 64: Were these filter paper disks?

Response 3: Yes, the disk diffusion assay was carried out by filter paper disks.

Point 4: Line 74: “peak” associated with absorption of specific wavelength?

Response 4: This was corrected to be “absorption peak of specific wavelength”.

Point 5: Line 87: Sate the type of column used here.

Response 5: The type of column used for is weak anion exchange chromatography DEAE-Sepharose Fast Flow Column.

Point 6: Line 93: Cut out instead of cut “off”

Response 6: It has been corrected according to the comments.

Point 7: Line 159: Delete “at”

Response 7: It has been corrected according to the comments.

Point 8: Line 200: Table1 and phylogenetic Figure, and description associated with them, belong to the Results section.

Response 8: Table1 and phylogenetic Figure, and description associated with them were moved to the Results section.

Point 9: I would suggest including at least one of your supplementary figures showing clearing zones in the main Results section so readers quickly get a clear idea of the assay being employed.

Response 9: The former Figure A1 which is a supplementary figure showing clearing zones is included in the main Results section.

Reviewer 2 Report

Materials and methods do not have adequate descriptions of how the experiments/protein purification were carried out

Flagellin is an abundant protein and could be a contaminant. Therefore, a flagellin mutant of the Bacillus strain is necessary to determine if the protein is required for antifungal activity. Additionally, testing the recombinantly expressed protein puriified from an expressing strain of E. coli for antifungal activity is also necessary.

other comments/corrections

L54, Please express centrifuge speed in xg rather than rpm

L55. ...were passed through 0.45 µm filters to remove debris. (?)

L58. In what buffer was the extract dialyzed. 

L59. At which point in the procedure was BaCl2 added?

L66. ...volume of buffer used to dissolve the proteins...

L80. Be more specific in describing which column was used. 

L84. Dialyzed in which buffer?

L87-90. Please specify which column was used and in which buffer dialysis was carried out. 

L97-98. Omit 'of it'

L99 and 102. Omit 'respectively'.

Author Response

Response to Reviewer 2 Comments

Point 1: Flagellin is an abundant protein and could be a contaminant. Therefore, a flagellin mutant of the Bacillus strain is necessary to determine if the protein is required for antifungal activity. Additionally, testing the recombinantly expressed protein purified from an expressing strain of E. coli for antifungal activity is also necessary.

Response 1: Thanks very much for these comments. The flagellin mutant and the recombinantly expressed protein were both excellent methods to determine antifungal activity. In this study, the antifungal activity was determined by measuring the diameters of inhibition zones of I. robusta growth for each protein. In bioresearch, this method is reliable to determine the antifungal activity for protein. Hence, we hold the opinion that the method used is appropriate in the current study. In the further study, we would consider to use corresponding methods to research about the antifungal activity.

Point 2: L54, Please express centrifuge speed in xg rather than rpm

Response 2: This has been corrected according to the reviewer’s comments, and it is 22,200 g.

Point 3: L55. ...were passed through 0.45 µm filters to remove debris. (?)

Response 3: This has been corrected according to the reviewer’s comments.

Point 4: L58. In what buffer was the extract dialyzed.

Response 4: There is no buffer used and the extract dialyzed in deionized water.

Point 5: L59. At which point in the procedure was BaCl2 added?

Response 5: BaCl2 was added after dialysis overnight at 4 oC and the corresponding text has been corrected to be “Then, BaCl2 was added to detect the white precipitate”.

Point 6: L66. ...volume of buffer used to dissolve the proteins...

Response 6: The volume of buffer used to dissolve the proteins was 80 μL.

Point 7: L80. Be more specific in describing which column was used.

Response 7: The column used was DEAE-Sepharose Fast Flow Column, which was produced by GE, American.

Point 8: L84. Dialyzed in which buffer?

Response 8: The protein collected from each eluted peak was dialyzed by deionized water.

Point 9: L87-90. Please specify which column was used and in which buffer dialysis was carried out. 

Response 9: The column used was DEAE-Sepharose Fast Flow Column which produced by GE, American. The buffer dialysis was buffer A which was deionized water.

Point 10: L97-98. Omit 'of it'

Response 10: ‘it’ has been omitted in L97-98.

Point 11: L99 and 102. Omit 'respectively'.

Response 11:respectivelyhas been omitted in L99 and 102.

Reviewer 3 Report

This manuscript by Yun Jiang et al. describes the purification and the characterization of a novel antifungal flagellin from the NJ13 strain of Bacillus methylotrophicus against Ilyonectria robusta.

The experiments are sound and described in sufficient details. While the study is not conclusive, it appears to pave the way to the discovery of the actual antifungal protein from strain NJ13.

Major:

The introduction is somewhat confusing and does not provide the reader with a clear and detailed picture of the background of the project. Please expand the introduction and make it more readable to people who are unfamiliar with the system. Fig. 1 is of poor quality. The band of the protein is difficult to visualize. Please try to improve the quality of the figure. Overall, for improving readability, I think that the paper would generally benefit from some further editing.

Minor:

Line 17: temperatures higher than 100 °C Line 17: did not tolerate organic solvents Line 19: Bacillus velezensis should be written in italics Line 30: please replace “which” with “that” Line 31: “…...., and immunity enhancement properties” Line 33: please replace “has been” with “is” Line 52: under pH 7.0 should be “at pH 7.0”. Also, please add the buffering agents. Line 59: whiteness? Do you mean white? Line 128: “…cells of E.coli DH5α” should become “E.coli DH5α cells” Line 159: please remove “at” Line177: please replace “cloned” with “transformed”…”transformed into E.coli DH5α cells”

Author Response

Response to Reviewer 3 Comments

Point 1: The introduction is somewhat confusing and does not provide the reader with a clear and detailed picture of the background of the project. Please expand the introduction and make it more readable to people who are unfamiliar with the system.

Response 1: Thanks very much for the comment. The introduction has been slightly expanded. To let the background of the project much more detail, the Ilyonectria robusta has been further introduced with “Ilyonectria robusta is an important pathogen for the rusty root rot of Panax ginseng and Panax quinquefolius”.

Point 2: Fig. 1 is of poor quality. The band of the protein is difficult to visualize. Please try to improve the quality of the figure.

Response 2: We have tried to improve the quality of Figure 1 through improve its colour depth.

Point 3: Line 17: temperatures higher than 100 °C.

Response 3: This has been changed according to the comments.

Point 4: Line 17: did not tolerate organic solvents.

Response 4: This has been changed according to the comments.

Point 5: Line 19: Bacillus velezensis should be written in italics

Response 5: This has been changed according to the comments.

Point 6: Line 30: please replace “which” with “that”

Response 6: This has been changed according to the comments.

Point 7: Line 31: “…...., and immunity enhancement properties”

Response 7: This has been changed according to the comments.

Point 8: L80. Be more specific

Response 8: This has been changed according to the comments.

Point 9: Line 33: please replace “has been” with “is”

Response 9: This has been changed according to the comments.

Point 10: Line 52: under pH 7.0 should be “at pH 7.0”. Also, please add the buffering agents.

Response 10: This has been changed according to the comments.

Point 11: Line 59: whiteness? Do you mean white?

Response 11: This has been changed according to the comments.

Point 12: Line 128: “…cells of E.coli DH5α” should become “E.coli DH5α cells”

Response 12: This has been changed according to the comments.

Point 13: Line 159: please remove “at”

Response 13: This has been changed according to the comments.

Point 14: Line177: please replace “cloned” with “transformed”…”transformed into E.coli DH5α cells”

Response 14: This has been changed according to the comments.

Reviewer 4 Report

Overall in submitted research article data is good enough for publication.

It requires minor revision before publication in "Microorganisms".

Comment:

Section 3.3: Author should cite these article to support the experimental findings:

RSC Adv., 2015, 5, 20115-20131.

Cell Biochem Biophys (2012) 62, 487-499.

Author Response

Response to Reviewer 4 Comments

Point 1: Section 3.3: Author should cite these articles to support the experimental findings:

RSC Adv., 2015, 5, 20115-20131.

Cell Biochem Biophys (2012) 62, 487-499.

Response 1: These two articles are cited in the section 3.3.

Round 2

Reviewer 2 Report

I do not think the authors
adequately addressed my critique. If the authors wish to be rigorous,
then they should perform one of the tests I suggested to confirm
their conclusion that flagellin possesses anti fungal activity. Also, the potency of the antifungal activity went from 31.2mg/L (L41) to ~20 mg/ml (Materials and Methods). This is a considerable loss of potency.  The problem here is that there is no data on the protein concentration used test antifungal activity. There is no MIC or LD50, data that one would normally expect to see in a paper about an antimicrobial agent.  Therefore, I do not have much enthusiasm for this report

Reviewer 3 Report

The manuscript has improved somewhat, although extensive language editing is still required prior to publication. For instance:

1) remove "high" from line 17

2) Line 31: please replace "immunity enhancement properties" with "immunity-enhancing properties"

etc.

I do believe that a thorough revision form a native speaker would improve the quality of the manuscript significantly.